# miRNAs in Cardiac Myxoma: New Pathologic Findings for Potential Therapeutic Opportunities

**DOI:** 10.3390/ijms23063309

**Published:** 2022-03-18

**Authors:** Antonio Nenna, Francesco Loreni, Omar Giacinto, Camilla Chello, Pierluigi Nappi, Massimo Chello, Francesco Nappi

**Affiliations:** 1Cardiac Surgery, Università Campus Bio-Medico di Roma, 00128 Rome, Italy; a.nenna@unicampus.it (A.N.); francesco.loreni@unicampus.it (F.L.); o.giacinto@unicampus.it (O.G.); m.chello@unicampus.it (M.C.); 2Integrated Biomedical Science and Bioethics, Università Campus Bio-Medico di Roma, 00128 Rome, Italy; c.chello@unicampus.it; 3Cardiology, Università degli Studi di Messina, 98122 Messina, Italy; pierluigi.nappi@unime.it; 4Cardiac Surgery, Centre Cardiologique du Nord de Saint Denis, 93200 Paris, France

**Keywords:** miRNA, myxoma, cardiac tumors, cardiac surgery

## Abstract

MicroRNAs (miRNAs) regulate gene expression at the post-transcriptional level, contributing to all major cellular processes. The importance of miRNAs in cardiac development, heart function, and valvular heart disease has been shown in recent years, and aberrant expression of miRNA has been reported in various malignancies, such as gastric cancer and breast cancer. Different from other fields of investigation, the role of miRNAs in cardiac tumors still remains difficult to interpret due to the scarcity publications and a lack of narrative focus on this topic. In this article, we summarize the available evidence on miRNAs and cardiac myxomas and propose new pathways for future research. miRNAs play a part in modifying the expression of cardiac transcription factors (miR-335-5p), increasing cell cycle trigger factors (miR-126-3p), interfering with ceramide synthesis (miR-320a), inducing apoptosis (miR-634 and miR-122), suppressing production of interleukins (miR-217), and reducing cell proliferation (miR-218). As such, they have complex and interconnected roles. At present, the study of the complete mechanistic control of miRNA remains a crucial issue, as proper understanding of signaling pathways is essential for the forecasting of therapeutic implications. Other types of cardiac tumors still lack adequate investigation with regard to miRNA. Further research should aim at investigating the causal relationship between different miRNAs and cell overgrowth, considering both myxoma and other histological types of cardiac tumors. We hope that this review will help in understanding this fascinating molecular approach.

## 1. miRNAs: The Clinical Concept

MicroRNAs (miRNAs) regulate gene expression at the post-transcriptional level, contributing to all major cellular processes. miRNAs have been recently recognized as biomarkers and possible therapeutic targets for the diagnosis and treatment of diseases [1]. A single miRNA can target many genes, affecting various gene expressions [2]. miRNAs impose a relatively weak effect on their target, showing that single mRNAs are targeted by multiple miRNAs, but manipulation of RNA using mimicry and antagonists has significant therapeutic potential for the treatment of a variety of diseases [3,4].

The importance of miRNAs in cardiac development and heart function in cases of ischemic disease has been shown in recent years [1,3,5,6,7,8,9,10,11]. Early stages of cardiac development are controlled by miR-1 and miR-133a, resulting in commitment of cardiac-specific muscle lineage from embryonic stem cells and mesodermal precursors [12]. In addition, gene regulation of miR-1 and miR-133a is implicated in the mesodermal and cardiac fate of pluripotent stem cells [12]. Other miRNAs, such as miR-208 and miR-499, are involved in the late cardiogenic stages, mediating differentiation of cardioblasts to cardiomyocytes and fast/slow muscle fiber specification. Cardiac automaticity is regulated by miR-1/133a, while miR-208/499 regulates expression of contractile proteins. In cardiac pathology, expression of cardiac miRNAs is markedly altered [1,3,5,6,7,8,9,10,11], leading to acute (ischemia-reperfusion and apoptosis) and chronic (fibrosis, hypertrophy, and remodeling) adverse effects. In acute myocardial infarction, circulating levels of cardiac miRNAs are significantly elevated, making them a promising marker for early diagnosis [5]. Moreover, the importance of miRNAs in heart valve disease was recently described [1]. The role of miRNAs in aortic stenosis seems to be better clarified, as miR-30b decreases in valve calcification through negative regulation exerted on osteoblastic differentiation [1]. Reduced levels of miR-141 guarantee osteoblastic calcification processes via increasing BMP-2 levels, and this differentiative process negatively affects valve elasticity by contributing to leaflet calcification [1].

Aberrant expression of miRNA has been reported in various malignancies, such as gastric cancer and breast cancer, and recent evidence has shown that circulating miRNAs can be regarded as reliable biomarkers [13,14,15,16]. Different from other fields of investigation, the role of miRNAs in cardiac tumors still remains difficult to be interpreted due to fragmented publications and a lack of narrative focuses on this topic. In this article, we summarize the available evidence on miRNAs and cardiac tumors, and define new pathways for future research. We tried to systematically search for all cardiac tumors, but miRNAs were investigated adequately only in cardiac myxomas, as shall be discussed.

## 2. miRNA and Cardiac Myxomas

Cardiac myxomas usually develop in the atrium and involve an acid-mucopolysaccharide-rich myxoid matrix with diffuse stromal cells, and they are the most common type of cardiac tumor. Despite being “oncologically” benign, they can have devasting clinical consequences as a result of embolization or heart valve functional stenosis. Recent gene expression and immunohistochemical studies have established that cardiac myxoma cells arise from multipotent mesenchymal cells, and a recent study clearly investigated this topic [17,18]. Cardiac myxoma cells have been shown to be multipotent c-kit-positive, CD45-negative, and CD31-negative, and they produce the characteristic gelatinous matrix, as well as the ability to be clonogenic, self-renewing, and sphere-forming [13,17,19]. Scalise et al. [17] found that myxoma cells have an miRNA network that closely resembles cardiac stem/progenitor cells from normal myocardium, except for some dysregulated miRNA (upregulation of miR-126-3p and downregulation of miR-335-5p) involved in cell growth, differentiation, and transformation. Those “myxoma-initiating cells” (c-kit-positive, CD45-negative, and CD31-negative, similar to multipotent cells), in cases of dysregulated miRNAs, can lead to clinical effects. As myxomas seem to be derived from multipotent mesenchymal cells [17], downregulation of miR-335 is known to cause de-repression of its target genes such as *RUNX2* [20]. This activates the reparative mesenchymal stem cell phenotypes, characterized by increased proliferative, migratory, and differentiation capacities, which can clinically result in myxoma formation [20,21]. Notably, this specific miRNA alteration can also be induced by interferon or other pro-inflammatory signals [20], and this might explain why remnants of myxoma tissue accidentally left by the first surgical removal can lead to recurrencies. In addition, miR-335 is known to have a pivotal role in the regulation of the reparative activities of mesenchymal stem cells [21] and might be deemed a marker of cell therapeutic potency in future studies.

Yan et al. recently found specific miRNA expression profiles in the serum of patients with cardiac myxoma [14]. Authors found four differentially expressed miRNAs: the expression levels of miR-320a and miR-1249-5p were upregulated while miR-634 and miR-6870-3p were downregulated compared to healthy controls. Through bioinformatic analysis, almost 500 target genes were controlled by miR-320a; most of them belonged to the bone morphogenetic protein signaling pathway, nicotinamide adenine dinucleotide pathway, and ceramide biosynthetic pathway. The clinical impact of miR-320a overexpression is reduced cell migration and induced growth arrest by targeting VEGF and MEF2D. MiR-634 is a known tumor suppressor in other histologic tumors, and this expression was lower in the serum of patients with myxoma, suggesting the pivotal role of miR-634.

Zhang et al. [22] found that miR-217 expression is downregulated in the tissues of patients with cardiac myxoma, that overexpression of miR-217 inhibits the proliferation and promotes the apoptosis of the primary myxomatous cells, and that the expression of miR-217 is inversely correlated with the expression of interleukin 6 (IL-6). These authors concluded that miR-217 can serve as an oncosuppressor in cardiac myxoma by directly targeting 3′-UTR of IL-6 gene, suggesting that the manipulation of miR-217 might be a potential therapeutic target. Similarly, Cao et al. [23] reported that downregulation of miR-218 promotes cell proliferation, supporting the hypothesis that miR-217 and miR-218 are potential targets for cardiac myxoma prevention and therapy due to their tumor suppression properties.

Qiu et al. [24] investigated the causal relationship among miRNA, transcription factors, and cardiac myxoma. Levels of PPARg were found to be inversely correlated with MEF2D, a biomarker of cardiac myxoma. Activation of PPARg inhibits MEF2D expression via upregulation of miR-122, through two different mechanisms: targeting the 3′-UTR of MEF2D to inhibit MEF2D expression or directly binding to the PPARg in a specific miR-122 promoter region. The clinical results of the experiment confirmed that miR-122-dependent downregulation of MEF2D by PPARg suppresses the proliferation of myxoma cells, suggesting that PPARg may exert its antiproliferative effects by negatively regulating the MEF2D. Therefore, upregulation of miR-122 and PPARg/miR-122/MEF2D signaling pathways may be a novel target for treatment of cardiac myxoma. In addition, MEF2 factor is an important promoter of miR-1 and plays a crucial role in the differentiation of cardiac embryonic stem cells to mesodermal cardiomyocytes [5]. MEF2 is overexpressed in cardiac myxoma, and, consequently, miR-1 is overproduced, thus enhancing the neoplastic differentiation of embryogenic cells [17].

Table 1 and Figure 1 summarize the available evidence about miRNAs and cardiac myxomas.

## 3. Biological Implications

Reactive oxygen species (ROS) affect many cellular functions and the redox balance between ROS and antioxidants contributes to cell physiology; an altered redox equilibrium leads to increased ROS production and oxidative damage. Several miRNAs are expressed in response to ROS to mediate oxidative stress. Conversely, oxidative stress may lead to the upregulation of the miRNAs that control buffer mechanisms against damage induced by ROS.

A recent review by Climent et al. [15] summarized the role of miRNAs on oxidative stress, concluding that miR-34a, miR-144, miR-421, miR-129, miR-181c, miR-16, miR-31, miR-155, miR-21, and miR-1/206 are crucial mediators. miRNAs counteract ROS both directly (by targeting oxidase and antioxidant enzymes, antioxidant genes, and their transcription factors, or mitochondrial genes) and indirectly, mainly by targeting genes involved in apoptosis and tumor-suppressor genes, and by interfering with pro-survival signaling pathways. miRNAs were recently shown to be potential targets and modulators of oxidative-stress-related pathways, especially considering nuclear factor erythroid 2 (NRF2), sirtuin (SIRT-1), and the NF-kB signaling cascade [11].

From a biological point of view, tumors are known to be related to oxidative stress. Even in the context of cardiac tumors, this relationship seems strong enough to support the hypothesis that oxidative stress enhances cell proliferation, which clinically results in cardiac tumors. As similar miRNAs can be shared between oxidative stress and tumors, the same therapeutic target can act on the proliferation pathway and on the inflammatory pathway.

## 4. Clinical Implications

Interleukin-6 (IL-6) is associated with an increased risk of adverse cardiovascular events, especially myocardial infarction, since IL-6 favors the onset of unstable plaque atherosclerosis or increases the inflammatory rate of already present plaques [25]. Consequently, in patients with cardiac myxoma, in addition the thromboembolic risk related to the mass, there is a strong possibility of rupture of unstable coronary plaques with consequent acute coronary syndromes. IL-6 increases the pro-inflammatory state, as shown by higher C-reactive protein levels, and might potentially impair cardiac performance [25], as the increase in inflammatory burden might enhance oxidative stress with enhanced ROS production, thus resulting in depressed ejection fraction.

From a clinical perspective, circulating miRNAs might be useful in the diagnosis of cardiac tumors as surrogate biomarkers, optimizing the diagnosis and treatment choices at an early stage [25]. Upregulation of miR-126 appears to be a marker of myxoma-initiating progenitor cells and could be a valuable serum biomarker in cases of early diagnosis and relapse. In addition, intraoperative administration of specific miR mimics or inhibitors can locally target cancer cells and may be a potential adjunctive therapy to surgical removal to avoid recurrence.

## 5. Future Therapeutic Implications

Surgical excision of cardiac myxomas remains the mainstay of treatment to avoid systemic or pulmonary embolization and detrimental complications. Therefore, the surgical approach should be recommended at diagnosis. However, recurrencies are not infrequent and might be related to incomplete surgical removal (e.g., leaving parts of myxoma cells in the interatrial septum) and/or specific genetic features, which are only partially known (e.g., Carney complex). The proposed miRNA-based treatments may play a crucial role in the near future to avoid recurrencies.

The efficacy of drugs must be titrated against their off-target effects, such as negative impacts on the complement system, toxicity related to the vector, or interference with innate immunity [2,26,27]. Although some antago-miRs have been used in clinical trials against the hepatitis C virus and in other conditions [28,29,30], no study has investigated their role in cardiac tumors. However, targeted therapy to miRNA is known to be not free from side effects. The inhibition of miR-320a suppresses the expansion of VEGF and other proliferation factors, potentially leading to left ventricular dysfunction and reduction in the ejection fraction, similarly to the known effects of monoclonal VEGF blockers, such as bevacizumab, pazopanib, sorafenib, and sunitinib [25]. Other side effects include hypertension, mediated by a reduced nitric oxide synthesis and its effects, and thromboembolic events, due to damage to the vascular endothelium, mediated by a reduction in angiogenic factors and increased platelet aggregation [25].

A relevant concern when using miRNA as a therapeutic target is the need to achieve stability and resistance to degradation enzymes. In current preclinical research, antisense nucleotides and vectors mimic miRNAs and prevent their degradation or transcriptional blockage. Potential approaches to improve stabilization are 2-O-methyl inclusion, 2-Fluorum inclusion, and a 3′-cholesterol tail [1]. In addition, improvements in the delivery systems to improve the kinetic parameters are under investigation, and liposomes and microbubbles appear extremely promising [1].

Specific miRNAs are currently being investigated as potential therapeutic targets, but the literature still lacks tailored publications. However, considering the pathophysiologic role of miRNAs in cardiac myxomas, different future therapeutic approaches may reliably be considered (Figure 2). Through the stimulation and inhibition of different miRNAs, the transformation of stem cells or fibroblasts into cardiomyocytes can be targeted, including all components of mature cardiomyocytes (such as sarcomeres and calcium flow) [5]. This can open new therapeutic avenues, especially in relation to hypofunctional myocardial tissues of ischemic heart disease or dilated cardiomyopathy [5].

## 6. Gaps in Evidence

At present, the study of the complete mechanistic control of miRNA remains a crucial issue, as proper understanding of signaling pathways is essential to forecast therapeutic implications [11,15]. The direct biological regulator is still not understood in most cases and future studies should carefully address those points.

Other types of cardiac tumors still lack adequate investigations in terms of miRNA. Therefore, tailored studies on other cardiac tumors are required. All reported studies have included patients with cardiac myxoma, as this is the most frequently observed cardiac neoplasm, and other tumors are less frequently encountered in clinical and surgical practice; consequently, there is little material for detailed histological evaluation and pathological studies.

In some settings, the production of ROS inhibits tumor growth, leading to oxidative stress and creating an environment highly unfavorable to cell growth; tumor proliferation decreases due to apoptosis induced by oxidative stress. This was observed with miR 506 through the NF-kB pathway and ROS production [15], and confirmation in the field of cardiac tumors would be extremely interesting. Can oxidative stress be beneficial by slowing the growth of cardiac tumors and promoting the apoptosis of cancer cells? Even if, theoretically, this can be harmful to the heart muscle, what would be the real clinical balance?

## 7. Conclusions

miRNAs are involved in cellular physiological reactions leading to cardiac myxomas, and multiple pathways are involved (Table 1). The results in this review might be useful for supporting further exploration of specific miRNAs. Even if pathological pioneering studies are promising, further studies are warranted to understand how miRNA signaling directly modulates the pathophysiology of cardiac tumors. Further research should aim at investigating the causal relationship between different miRNAs and cell overgrowth, considering both myxomas and other histological types of cardiac tumors. Surgical removal remains the mainstay of treatment, but the role of miRNA-based treatment might play a crucial role in avoiding recurrencies in the near future. We hope that this review will help in understanding this fascinating molecular approach.

## Figures and Tables

**Figure 1 ijms-23-03309-f001:**
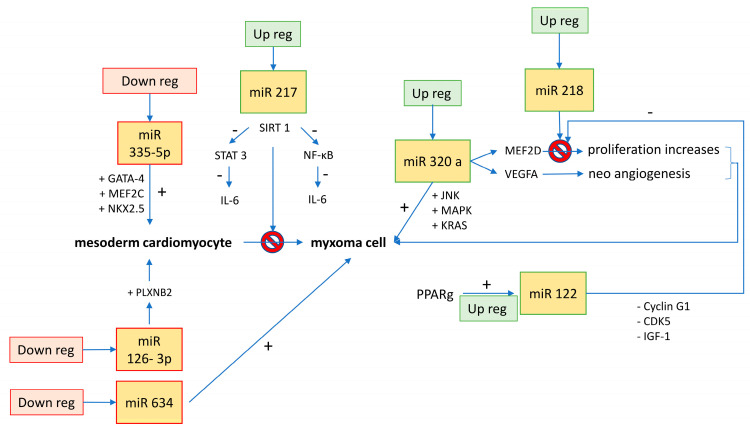
Regulation of miRNAs in cell differentiation from stem cells to myxoma cells.

**Figure 2 ijms-23-03309-f002:**
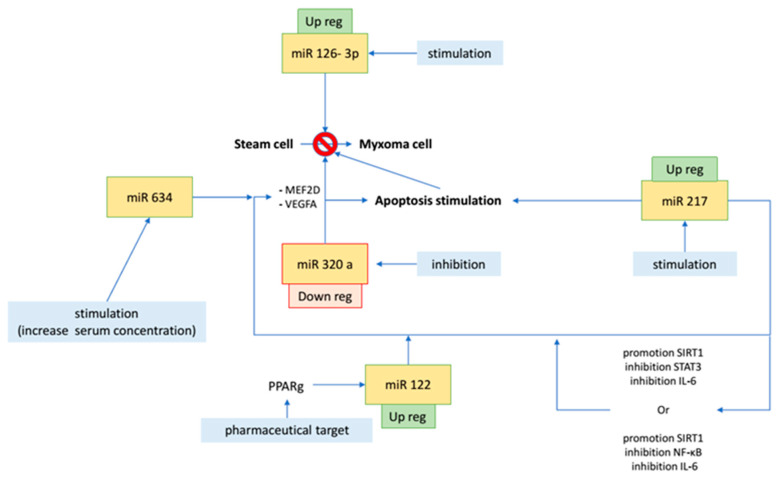
Potential therapeutic targets for cardiac myxomas.

**Table 1 ijms-23-03309-t001:** Summary of miRNAs implicated in cardiac myxomas.

MiRNAs	Biological Effect	Biological Control	Macroscopic Effect	Potential Therapeutic Implication	Reference
miR-335-5p	downregulation increases the expression of cardiac transcription factors (GATA-4, MEF2C, and NKX2.5)	?	differentiation from embryonic stem cell to mesoderm cardiomyocyte	inhibition could enhance new myocyte formation from stem and progenitor cells	[17]
miR-126-3p	downregulation increases cell cycle trigger factors such as PLXNB2	?	cell growth	delay in cell overgrowth	[17]
miR-320a	overexpression induces MEF2D, VEGFA; interferes with the de novo biosynthesis of ceramide (apoptosis, cell differentiation, and proliferation)	?	proliferation, cell invasion, apoptosis, and size increase (JNK, MAPK, KRAS); lipotoxicity in diabetic heart	ceramide-mediated apoptotic signaling pathways have been considered targets for anticancer therapies; inhibiting miR-320a unlocks the production of ceramides, promotes apoptosis, and inhibits MEF2D/VEGFA blocking cell proliferation and neoangiogenesis	[3,14]
miR-634	?	?	oncosuppressor; inhibits cell proliferation and induces apoptosis	increases chemotherapy-induced cytotoxicity	[14]
miR-217	suppresses production of IL-6 through different pathways: (1) promotion of SIRT-1 and inhibition of STAT-3 (cell cycle pathway); (2) promotion of SIRT-1 and inhibition of NF-kB (inflammatory pathway)	controlled by Ehmt1/Ehmt2 histone methylation (for cardiac hypertrophy)	inhibits proliferation and promotes apoptosis; implicated in cardiac hypertrophy	pharmacologically reduces the concentration of IL-6 through miR-217 stimulation	[19,22]
miR-122	through PPARg pathway, promotion of miR-122 inhibits MEF2D	PPARg	blocks proliferation and induction of apoptosis or, in any case, blocks oncogenic factors (cyclin G1, CDK5, and IGF-1)	stimulation of PPARg to block MEF2D	[24]
miR-218	MEF2D inhibitor	Dnmt3b DNA methylation	tumor suppressor when overexpressed (pro-apoptotic and reduces proliferation)	reduces cell proliferation through stimulation of miR-218/MEF2D pathway	[14,19,23]

## Data Availability

Not applicable.

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
