# Peer review of "miRNAs in Cardiac Myxoma: New Pathologic Findings for Potential Therapeutic Opportunities"

_ijms, 2022, doi:10.3390/ijms23063309_

Round 1

Reviewer 1 Report

The review by Nenna et al, focused in the scientific knowledge on alteration in miRNAs and the biology of myxomas, is timely and interesting, well organized and written; it is a good review, in a fragmented body of  knowledge in an infrequent benign tumor.

  1. As myxomas seem to be derived from multipotent mesenchymal cells (Scalise et al., 2020), conclusions on miR-335 in mesenchymal stem cells could be of relevance for the described model (Tome et al., 2011; 2014), and should be considered, and perhaps discussed.
  2. Along the review, in several occasions, it is referred to the potential therapeutic opportunity of several miRNAs. For the general public I would recommend to explain how can it been designed to be combined with the state-of -the art treatment. I guess that the surgical excision should be unavoidable, and the proposed treatments could help to avoid the frequent recurrences. Is it the case?
  3. miR-1 has also demonstrated activity in adult cardiac progenitors (Izarra et al., 2017).
  4. Figure 1 indicates some points that could not be clear enough for the reader. The effect of miR-355-5p (DownReg) is only indicated on the embryonic stem cell to mesodermal cardiomyocyte transition. Can we understand that in some patients the origin of the tumor could me initiated during embryonic development? Is there any evidence on that? Where would be located the multipotent mesenchymal cells or cardiac progenitor cells?

Minor points

Pag 1, lane 20             MiRNA     miRNA

Author Response

Reviewer #1

The review by Nenna et al, focused in the scientific knowledge on alteration in miRNAs and the biology of myxomas, is timely and interesting, well organized and written; it is a good review, in a fragmented body of knowledge in an infrequent benign tumor.

Dear reviewer, thank you for your comments and your appreciation. Manuscript has been revised according to your suggestions.

  1. As myxomas seem to be derived from multipotent mesenchymal cells (Scalise et al., 2020), conclusions on miR-335 in mesenchymal stem cells could be of relevance for the described model (Tome et al., 2011; 2014), and should be considered, and perhaps discussed.

Thank you for your comment. We included the cited references (10.1038/cdd.2010.167 and 10.1002/stem.1699) and discussed them in the miR-335 section. Biological implications and therapeutic opportunities have been pointed out accordingly.

  1. Along the review, in several occasions, it is referred to the potential therapeutic opportunity of several miRNAs. For the general public I would recommend to explain how can it been designed to be combined with the state-of -the art treatment. I guess that the surgical excision should be unavoidable, and the proposed treatments could help to avoid the frequent recurrences. Is it the case?

Thank you for your comment. We completely agree with reviewer’s point. The paragraph “therapeutic implications” and the conclusions have been revised accordingly.

  1. miR-1 has also demonstrated activity in adult cardiac progenitors (Izarra et al., 2017).

Thank you for your comment. We included the cited reference (10.1002/term.1977).

  1. Figure 1 indicates some points that could not be clear enough for the reader. The effect of miR-355-5p (DownReg) is only indicated on the embryonic stem cell to mesodermal cardiomyocyte transition. Can we understand that in some patients the origin of the tumor could me initiated during embryonic development? Is there any evidence on that? Where would be located the multipotent mesenchymal cells or cardiac progenitor cells?

Thank you for your comment. We sincerely apologize for our mistake and the Figure 1 has been revised accordingly. To avoid ambiguity, embryonic stem cells have been removed. Reference for this mechanism is Scalise et al. (2020).

  1. Pag 1, line 20 – MiRNA / miRNA

Thank you for your comment. Spelling has been corrected accordingly.

Reviewer 2 Report

The article presented by Nenna A et al entitled: “MiRNAs and cardiac tumors: new pathologic findings, potential therapeutic opportunities” is an interesting topic.

This review helps to better understand the role of the miRNAs in cardiac tumor but in my opinion some major revisions are necessary.

The authors speak about cardiac tumors but in the article, they only discuss about myxoma.

 I agree that myxoma is the main type of cardiac cancer but both the title than the aim of the article must be modified indicating that you will only analyze the myxoma as it is the prevalent heart cancer or necessarily the other types of cancer need to be discussed.

Moreover, the paragraph “ 5. Therapeutic implications” must be implemented with more information because those provided are very limited and superficial.

Author Response

Reviewer #2

The article presented by Nenna A et al entitled: “MiRNAs and cardiac tumors: new pathologic findings, potential therapeutic opportunities” is an interesting topic. This review helps to better understand the role of the miRNAs in cardiac tumor but in my opinion some major revisions are necessary.

Dear reviewer, thank you for your comments and your appreciation. Manuscript has been revised according to your suggestions.

The authors speak about cardiac tumors but in the article, they only discuss about myxoma. I agree that myxoma is the main type of cardiac cancer but both the title than the aim of the article must be modified indicating that you will only analyze the myxoma as it is the prevalent heart cancer or necessarily the other types of cancer need to be discussed.

Thank you for your comment. As described at the bottom of paragraph 1, “This article intends to summarize the available evidence on miRNAs and cardiac tumors, trying to define new pathways for future research. We tried to systematically search for all cardiac tumors, but miRNAs were investigated adequately only in cardiac myxomas, which will be discussed.”. However, we agree with reviewer’s point. We revised the title accordingly and we focused our manuscript on myxomas in both abstract and text. As an additional comment, authors were not able to find any significant contribution to this field in other types of cardiac tumors. Table 1 has been revised accordingly.

Moreover, the paragraph “ 5. Therapeutic implications” must be implemented with more information because those provided are very limited and superficial.

Thank you for your comment. We pointed out the importance of surgical excision and the role of miRNAs as a future therapeutic approach to avoid recurrencies. We revised the structure of this paragraph, but literature lacks tailored studies about this topic and our conclusion in this paragraph are speculative and only hypothesis-generating. A proper study design should include patients /animals undergoing surgical removal and then miRNA-specific therapy to avoid recurrencies, and we were not able to find similar studies.

Round 2

Reviewer 2 Report

Accept in present form